# Successful and Emerging Cyberbullying Prevention Programs: A Narrative Review of Seventeen Interventions Applied Worldwide

Sohni Siddiqui * and Anja Schultze-Krumbholz

Department of Educational Psychology, Technische Universität Berlin, 10587 Berlin, Germany;
anja.schultze-krumbholz@tu-berlin.de
* Correspondence: s.zahid@campus.tu-berlin.de

**Abstract:** The advent of the internet has channeled more online-related tasks into our lives and they have become a pre-requisite. One of the concerns with high internet usage is the multiplication of cyber-associated risky behaviors such as cyber aggression and/or cyberbullying. Cyberbullying is an emerging issue that needs immediate attention from many stakeholders. The aim of this study is to review existing successful and emerging interventions designed to prevent cyberbullying by engaging individuals through teacher professional development and adopting a whole-school approach. The review presents the strengths and limitations of the programs and suggestions to improve existing interventions. Preparing interventions with a strong theoretical framework, integrating the application of theories in interventions, promoting proactive and reactive strategies in combination, beginning with baseline needs assessment surveys, reducing time on digital devices and the digital divide among parents and children, promoting the concepts of lead trainer, peer trainer, and hot spots, focusing on physical activity, and use of landmarks are some of the recommendations proposed by the authors. In addition to face-to-face intervention sessions, it is suggested to update existing intervention programs with games and apps and to evaluate this combination.

**Keywords:** cyberbullying; anti-bullying programs; teacher professional development; individualized training; whole-school intervention



## 1. Introduction

The advent of the internet has channeled more online-related tasks into our lives and they have become a pre-requisite. One of the concerns with high internet usage is the multiplication of cyber-associated risky behaviors such as cyber aggression. The term "cyberbullying" is defined as the deliberate infliction of harm using electronic methods, targeting individuals or groups of people, regardless of their age, who perceive such actions as offensive, derogatory, harmful, or unwanted [1]. Despite efforts and interventions, cyberbullying and hate messaging is still on the rise worldwide [2,3]. Many interventions deal with traditional/face-to-face/offline school bullying and are modified for cyberbullying issues on the basis of the similarities shared by both types of bullying behavior, such as unjustified aggression, being based on a power imbalance, and persevering over time [4,5]. Despite similarities, there are also differences, as stated by Smith (2012), such as cyberbullying requiring technological expertise, the unidentified perpetrator does not usually see the victim's reaction instantly, the roles of bystanders are more complex, and there are differences in intentions [6]. It is difficult to protect oneself against cyberbullying, as nasty messages or content can be sent to mobile phones, computers, or social media anytime and anywhere within seconds [7]. Berne et al. (2019) reported that experiencing cyberbullying as a victim results in negative emotions, including anger, anxiety, fear, and shame [8]. Furthermore, victims of cyberbullying tend to exhibit more somatic symptoms,

such as headaches and stomachaches, than their peers who have not experienced cyberbullying. Cyber victims also tend to report lower satisfaction with their overall appearance, body image, and weight than non-cyber victims. Additionally, it was found that female victims of cyberbullying reported a more negative perception of their general appearance in comparison to male victims of cyberbullying [8]. Considering the global prevalence and detrimental consequences of cyberbullying, researchers have proposed preventive and interventional approaches to discourage children and adolescents from cyberbullying [9]. Additionally, strategies have been developed to help cyber victims manage adverse effects. These prevention methods also encompass school-based interventions, involving the training of teachers and staff members to enhance the overall school environment and foster a conducive learning atmosphere [9].

The meta-analysis conducted by Gaffney et al. (2019) indicated that cyberbullying intervention programs have proven effective in reducing both cyberbullying perpetration and victimization [10]. However, a recent systematic review by Torgal et al. (2023) demonstrated that the overall treatment effects of school-based cyberbullying intervention programs were not statistically significant [11]. These findings highlight the importance of conducting more comprehensive evaluations of cyberbullying intervention programs to identify the factors that contribute to the overall success rate of these programs.

The purpose of this study is to review and evaluate evidence-based individual training, teacher professional development programs, and whole-school anti-bullying interventions to control cyberbullying. This review examines the commonalities and distinctions among various cyberbullying intervention programs, considering factors such as the theoretical framework employed, content, activities, duration, inclusion of baseline needs assessment, allowing participants to adapt their learning, utilization of computer games and online resources as intervention tools, engagement of peers, and the incorporation of diverse incentives to motivate participants. The authors also aim to determine and compare the strengths and shortcomings of the interventions and to make recommendations for improvements. For this narrative review, the first author conducted a Google search using keywords like "school-based cyberbullying interventions", "individual training for cyber control", and "interventions through teacher professional development", etc. She specifically focused on interventions that demonstrated some level of effectiveness, ultimately identifying approximately 65 interventions that had been implemented and evaluated at the time of review. To ensure the credibility of the interventions, only those with multiple evidence-based research publications were included. After studying 17 of these interventions, the first author observed that the content and activities of subsequent interventions were largely repetitive. This led her to conclude the study at the 17th intervention.

The review classifies the interventions into three categories: engaging individuals in cyberbullying interventions, implementing teacher professional development, and adopting a comprehensive whole-school approach. The interventions aimed at curbing cyberbullying through teacher professional development were developed in response to research findings. These findings indicated that teachers held contradictory views and assumptions about cyberbullying, but through professional development, their perspectives were brought into alignment [12]. Furthermore, it was discovered that a strong bond between students and teachers can decrease the likelihood of negative bullying outcomes [13]. Another benefit of involving teachers was the cost-effectiveness of teacher-led interventions in comparison to externally delivered programs, making them a viable choice, particularly in low- to middle-income countries [14]. Additionally, the constant presence of teachers in classrooms throughout the academic year allowed students to seek help whenever needed in cases of bullying or victimization [15]. Conversely, offering direct support to individual students or victims was based on the premise that this approach eliminated the necessity for victimized children to reach out to adults. This safeguarded youngsters from harboring suspicions and facing accusations from adults, empowering them to independently confront bullying [16]. Moreover, individual interventions targeting cyberbullying spared victims from the ineffective and impractical remedies often offered by adults [17]. Comprehensive school-wide

strategies were deemed more effective than individual interventions or isolated teacher professional development, largely because they involved the entire community and were perceived as highly successful [18]. To cultivate a unified community, it is imperative that parents, educators, and the community as a whole receive training on adolescent matters [19]. This training equipped them with the knowledge and skills to raise awareness and provide coping strategies, enabling them to nurture young individuals' self-esteem and establish a foundation of trust [19]. Additionally, these interventions mitigated the digital divide between parents and children [20]. Table 1 and the following sections provide detailed information about the interventions examined.

## 2. Interventions Designed for Professional Development of Teachers

Teacher professional development programs are based on the premise that teachers are the primary agents capable of modifying the school environment by using their competencies to reduce bullying and victimization [21]. Teacher-led intervention programs are deemed effective and cost-efficient, particularly in low- to middle-income nations, when compared to programs administered by external psychologists [14]. A review of the existing literature suggests that enhancing teacher professional development can effectively tackle behaviors that hinder or interrupt student learning, as educators require the essential abilities to handle challenging behaviors and issues related to antisocial conduct [15,22,23]. In addition, teacher-led interventions have demonstrated positive outcomes in various aspects of education, such as fostering inclusive practices, supporting children with special needs [24,25], effectively addressing challenging behaviors [23], minimizing gender discrimination, and promoting equality, among others [26]. This indicates that professional development for teachers holds immense potential not only to impact the educational environment but also to indirectly influence the community and social practices.

The literature discusses many teacher-led interventions to control cyberbullying issues, but these programs have shown contrasting results [27]. This study focused on evidence-based, successfully evaluated anti-cyberbullying programs and summarized their content, duration, similarities, strengths, limitations, and success rates. Table 1 provides an overview and the main characteristics of the programs reviewed in the present publication.

### 2.1. P.E.A.C.E (Preparation, Education, Action, Coping, Evaluation) Pack

P.E.A.C.E Pack is a teacher-led anti-bullying program [28] developed by Phillip T. Slee in Australian schools [29], but it has shown a significant reduction in victimization in different school settings across the globe [28,30–32]. The content of the intervention is based on social constructivism to professionally develop teachers' skills in order to teach students positive attitudes and build social-emotional skills such as kindness, sensitivity, optimism, adaptability, conflict resolution strategies, productive coping skills, and positive emotions [33]. Teachers receive a manual with step-by-step instructions on how to implement the program in classrooms [28]. General activities include teacher-led discussion sessions and group activities aimed at creating conducive relationships [28]. A video presentation with a hypothetical story of a victim is used to create empathy and serves as input for a group discussion to reflect and recognize the consequences of insults and humiliation [28]. To reinforce the concept of cyberbullying and develop a constructive self-image, optimistic attitude, and conflict resolution skills, several short videos about cyberbullying are used to reflect on the victim's feelings and how others can support the victim [28]. An investigation encompassing Australian students spanning an age range of 5.4 to 13.5 years was conducted to assess the program's efficacy in mitigating bullying [33]. The intervention was observed to be successful in decreasing the prevalence of bullying ($p < 0.01$) [33]. The program has achieved favorable outcomes in cross-cultural settings, demonstrating its adaptability to diverse cultures through a wide range of research studies [33–35], and it has the potential to be adapted for more diversified cultural settings.

*2.2. ViSC (Viennese Social Competence Program)*

Based on the socio-ecological model, ViSC aims to prevent school-based traditional bullying as well as cyberbullying and cybervictimization [36]. The program is implemented via several in-school teacher workshops and a class project for students [21]. Gradinger and Strohmeier (2018) elaborated the details of the intervention and explained that the program starts with training teachers to recognize and handle bullying and introduce preventative measures and interventions at the school [36].

Later, teachers are trained to disseminate the philosophy and materials of the ViSC class project to students and train them in whole-school anti-bullying culture [36]. ViSC coaches train the teachers and usually start the sessions with brainstorming discussions to analyze and connect their previous knowledge, ideas, and beliefs about the phenomenon to the current situation [36]. Hypothetical bullying case studies are used to help teachers identify bullying and develop an understanding of how to overcome or intervene in such cases [36]. Teachers are equipped with communication skills needed to empower victims, counsel bullies, and communicate with guardians in case parental involvement is required [36]. Activities such as role playing, discussions, worksheets, small group activities, interactive games, etc. are used to train children [36].

This intervention has been shown to be successful with sustainable results in different countries, and it can be applied to low- to middle-income countries. However, training ViSC coaches can add further costs to the project and a financial analysis is necessary before applying the intervention to underprivileged economies. In a study conducted in Turkey by Doğan et al. (2017), students (mean age = 10.06 years) showed significant reductions in post-victimization and perpetration ($p < 0.001$) [37]. Another study by Solomontos-Kountouri et al. (2016) in Cyprus concluded that the program was more effective for grade 7 students than grade 8 students, but no effectiveness against cyberbullying was found in grade 7 students ($p > 0.001$) [38]. Additionally, an evaluation of an ultra-short, cost-effective version of the program implemented on students (mean age 13.28 years) in Kosovo by Arënliu et al. (2020) demonstrated a significant reduction in physical victimization ($p = 0.023$), although the cyberbullying reduction was not significant [39].

*2.3. Relationships to Grow (RPC)*

The RPC is a short intervention based on the idea of resilience and social exclusion designed for educators to prevent cyberbullying by fostering positive relationships among students [40]. The training's content is somewhat similar to other successful and effective strategies used in other anti-bullying programs, such as Asegúrate, KiVa, ConRed, Media Heroes, etc. The aim of RPC is to disperse knowledge about cyberbullying and increase proactive coping strategies against the phenomenon [41]. The program content is based on digital literacy, raising awareness about cyberbullying and coping skills, fostering collaboration and social skills, empathy, and sensitization training [41]. The RPC is a brief intervention program conducted exclusively by teachers, consisting of 6 h of teacher training, four activities implemented by teachers in their classrooms during school hours (each activity lasting 1.5–2 h), and 1 h of teacher supervision provided by expert psychologists [41]. It is a short-term intervention for teacher professional development that can be applied to underprivileged economies to increase awareness and initiate steps against cyberbullying. The program implemented on 6th–8th grade Italian students yielded positive outcomes by enhancing students' understanding of cyberbullying and its related risk factors, as well as improving their coping skills ($p < 0.001$) [41]. However, further research is needed to confirm its effectiveness in reducing cyber victimization and perpetration [41]. These findings demonstrate the program's potential in addressing cyber risky behaviors, but further research is necessary, including more comprehensive versions of the program and detailed studies, to expand understanding of the application of this intervention.

*2.4. Media Heroes*

Media Heroes is based on the Theory of Planned Behavior for cyberbullying prevention [42]. Media Heroes is a psychosocial intervention that aims to create and improve empathy/awareness among youth by involving their teachers and parents [42]. The program aims to raise awareness of cyberbullying and the consequences of perpetration and victimization, as well as to teach skills for safe use of the internet and supporting positive responses from bystanders [43]. Initially, teachers are trained by psychologists with use of a training manual to implement their skills within the existing school curriculum [44]. Part of the training program is to keep educators informed about student media use/activities that support and encourage healthy discussions between teachers and students beyond the context of the intervention. The program is delivered through a variety of techniques, including peer-to-peer tutoring, watching videos, presentations, role playing, discussions, debates, etc. [45]. To raise awareness among parents, students prepare a short "workshop" in which they present their perspective and information using a variety of activities such as role play, discussion, posters, and flyers [46]. It addresses teacher professional development in the curriculum and is also available in a short version (4 sessions for 90 min), which makes it economical for low- to middle-income countries; however, a short-term program still needs to be further developed to produce more profound results in reducing cyberbullying and victimization [41]. A study was conducted by Schultze-Krumbholz et al. (2015) on 722 high school students (mean age = 13.36 years) to determine the effectiveness of Media Heroes in reducing cyberbullying [43]. While the short-term intervention did not significantly affect cyberbullying change ($p$ = 0.113), the long-term intervention was effective ($p$ = 0.004) [43].

*2.5. Asegúrate*

The Asegúrate program aims to professionally equip educators to use a full range of resources to combat cyberbullying and its consequences [47]. The activities of the Asegúrate program are based on three important psychosocial theories: the Theory of Normative Social Behavior, the principles of constructivist methodology, and the development of self-regulatory skills [48]. Activities based on the principles of constructivism reflect pre-existing ideology and beliefs about the phenomenon, and sessions are developed to first understand participants' beliefs through brainstorming sessions [48]. Self-regulation skills are developed through reflection exercises to improve students' metacognitive skills and strategic learning [48,49].

Del Rey et al. (2019) provided details of the program, which is guided by a manual, audiovisual posters, stickers, and bookmarks to create awareness of cyberbullying and its consequences, the role of social networks in communication, maintaining anonymity in online activities, safe online practices, cyber gossip, sexting, cyber etiquette, and digital citizenship. Access to additional resources and information, such as reading materials, video links, etc., are also provided to teachers to further build their skills [48]. Teachers are also trained to engage students and families in training and awareness campaigns using a variety of methods [48]. The activities carried out by the teachers for the students are planned systematically, considering the students' activities and reflection on these activities and their consequences [48]. The final session ends with an individual and/or group of students making a commitment to actively address cyberbullying, which is one of the unique features of this project [47]. Researchers using a quasi-experimental approach conducted a study involving 479 secondary students (mean age = 13.83 years) that encompassed two measurements over time. The outcomes indicated that cyber aggression escalated without intervention, but decreased when the intervention was implemented ($p$ = 0.011, $\eta^2$ = 0.151) [47].

## 3. Bullying Prevention by Training Individuals

Adolescent victims often hesitate to confide in adults about their problems. They highly value privacy and seek anonymous assistance through peer support [50,51]. There

are reports suggesting that young individuals avoid involving adults in victimization matters due to a lack of trust and fear of being blamed [16]. Frequently, children choose not to disclose incidents of bullying because they feel ashamed of being a target [16]. Despite encouragement, many bullied students refrain from disclosing or seeking adult intervention in their difficulties [52] and typically reach out to their peers for assistance. Sulkowski et al. (2014) concluded that in most cases of reporting to adults (about 2 out of 3), the strategies offered were either unsuccessful/unhelpful or made the situation worse [17]. Based on this conclusion, the following interventions aim to practically empower individuals to cope with bullying. During individual training sessions, victims receive instruction on how to advocate for themselves and protect themselves against continued bullying without relying on teachers, peers, or parents. This empowers them to avoid shame and prevents labeling as victims. They do not need to disclose their personal struggles and secrets to their acquaintances, which maintains their self-identity and self-esteem. However, lack of adult supervision and loss of motivation over time lead to high dropout rates and call into question the effectiveness of such programs, which aim to train individuals to cope with bullying on their own.

### 3.1. Stand-Alone (Stop Bullies Online)

Stop Bullies Online/Stop Online Bullies is one of the applications in the Stand-Alone program and is designed for cyber victims (12–15 years) to empower themselves against bullying using an online, computer-tailored intervention [53]. The contents of the program were developed using multiple strategies, such as information gathered from a literature review, data collected through a Delphi study among experts, focus group interviews with the target group, and successful elements from a previously tested anti-bullying program [54]. The program is delivered through web-based counselling, with the first part changing participants' behaviors through reflection and debate, replacing irrational thoughts with impartial and balanced reasoning [53]. The second part provides awareness of cyberbullying and its consequences, bystander roles, and information on effective coping strategies to resist bullying [53]. Finally, individuals are trained to avoid risky online behaviors and safely use the internet and mobile phones [53]. The distinctive feature of the program is that content delivery is based on intervention mapping in which every component of the advice is personalized to the participants' personal characteristics (i.e., their self-efficacy, the way they cope with problems, and (ir)rational thoughts). This is considered a useful strategy for needs assessment and to find effective solutions [55]. Direct student involvement and lack of adult supervision can be ambiguous, as students tend to lose motivation or discipline and sometimes become disinterested or reluctant to be part of the program. Further research is recommended to confirm this program's effectiveness.

### 3.2. Cyberbullying Sensitization Program

The Cyberbullying Sensitization Program was created, executed, and assessed in an Indian region with the premise that raising awareness and sensitization about bullying would be beneficial. The program focuses on equipping individuals with the necessary skills to protect themselves from cyberbullying and promote positive online behaviors [56]. This intervention aims to raise children's awareness of cyberbullying as a coping strategy [56]. The program content creates awareness among youth and provides information about online bullying and its consequences, different types of bullying, threats, safety, and strategies to save oneself and others from unsafe risky online activities [56]. Targeted group discussions with adolescents about online bullying were the main methods used to develop this program, followed by an extensive literature review [57]. The intervention was developed by involving youth who were either perpetrators or victims of bullying and collecting their ideas, which can be considered a strength of the program [56]. The content validity of CBSP was determined by 14 experts in the fields of education, ICT, and law [56].

The intensive program consists of exercises designed to create awareness regarding the online world, activities, motivational reasons, strategies for dealing with cyberbullying,

identifying bullying, the role and responsibility of bystanders, and recommendations for online safety [56]. Some of the common activities used to teach the content are role playing, case studies, video presentations, and creative writing [19]. The strategies and content are similar to most Western interventions. The qualitative study conducted on 14–15-year-old students yielded encouraging outcomes by enhancing students' awareness of cyberbullying and fostering positive online behaviors [56]. This project provides a good start for under-resourced countries to take initiative against bullying.

### 3.3. Informational Motivational Behavioral Skills (IMB)

Information Motivational Behavioral Skills (IMB) is founded on a strong theoretical framework and based on the health behavior change framework proposed by Fisher and Fisher (1992) [58]. The framework is based on the enrichment of cognitive skills (information and motivation) and behavior (such as improvement in practical skills) [59]. The IMB model is not only focused on providing information about cyberbullying and its consequences, but also on motivating individuals to develop positive behaviors to prevent cyberbullying and develop practical skills that can help control online risk factors [59]. The unique feature of this intervention is that it is tailored to the individual's current information, motivation, and behavioral skills. This makes it particularly relevant and effective for specific characteristics and contexts [60]. Discussions, card-making group activities, sharing experiences, reflecting on self-practices, and developing a problem-solving attitude are targeted activities in the intervention [59]. A follow-up questionnaire is also utilized to evaluate the effectiveness of the program. The components and goals of this intervention are similar to those of other interventions that focus on disseminating information, engaging in skill-building activities, recognizing bystander responsibility toward victims and perpetrators, and creating a positive classroom environment. The research findings on 13–16-year-old South African female students revealed that the intervention group exhibited a higher perception of online risks ($p = 0.001$, $\eta^2 = 0.07$) [59]. This indicated the effectiveness of the intervention in enhancing online risk perception, which is a crucial factor in promoting positive behavioral change, with a small effect size [59]. Researchers have recommended that the program's effectiveness can be improved by applying it in long-term studies.

### 3.4. Prev@cib Anti-Bullying Program

Prev@cib is based on three theoretical frameworks: the ecological model, Empowerment Theory, and the personal and social responsibility model [61]. The ecological model is the most studied theory used in cyberbullying interventions to combat bullying by involving not only the individual, but everyone in the environment who can contribute to the intervention [62]. Empowerment Theory focuses on the empowerment of individuals and their resources to enable youth to take control of their lives in both virtual and school settings [63]. The personal and social responsibility model, as the name indicates, encourages shared responsibility in problem-solving to achieve greater involvement of adolescents in creating a bullying-free culture [64]. Prev@cib consists of three modules that start with raising awareness regarding cyberbullying and its consequences, with a special focus on sexting and cyber grooming [61]. In the second phase of the program, sensitization, empathy towards victims, and understanding social responsibility are highlighted through different activities [61]. The Prev@cib program is essentially designed to educate students, but teachers' opinions are also sought for successful implementation [61]. The study findings by Ortega-Barón et al. (2019) on secondary students (mean age= 13.58 years) revealed noteworthy reductions in bullying, victimization, cyberbullying, and cybervictimization within the experimental group, as opposed to the control group [61]. The findings revealed that in the control group, cyberbullying exhibited a consistent level, whereas in the experimental group, it demonstrated a reduction ($p < 0.01$) with a modest effect size of $\eta^2 = 0.05$. As for cybervictimization, a minor increase was noted in the control group, whereas the experimental group experienced a decrease ($p < 0.001$) with a slight effect size of $\eta^2 = 0.04$ [61].

These results demonstrated the effectiveness of the Prev@cib program in reducing bullying and cyberbullying [61]. The program has produced successful results in Spain; however, more research-based studies are needed in the national and international context to further elaborate on the outcomes of this program.

## 4. The Whole-School Approach

The whole-school approach is based on the belief that bullying is a systemic problem and that interventions need to focus on the whole-school context, rather than individual bullies and victims [18,65]. Interventions based on the whole-school approach seek to effectively prevent bullying and promote safe, supportive, responsible, engaged, and thriving school communities through ongoing school climate development and reform [66]. However, there are certain limitations associated with a whole-school approach, such as high cost, time commitment, and the need for a high level of support from schools as well as full parental cooperation [67].

### 4.1. KiVa Anti-Bullying Program

One of the most successful school-based, teacher-led interventions aims to raise awareness of bystander responsibility in promoting bullying, increase sensitivity to victims, and help individuals use strategies to support themselves and victims against bullying [68]. These goals are achieved by engaging children and adolescents, with the help of teachers, in activities such as discussions, presentations, illustrations in the forms of pictures, figures and characters to depict different aspects of cyberbullying, short films, assignments with various learning-by-doing exercises, and a computer game in which students practice new skills against bullying in a virtual environment, with the goal of improving students' understanding and knowledge of cyberbullying [68].

When bullying incidents are reported, KiVa-trained school members engage the individuals and groups involved in bullying in conversations to counsel them and correct their behaviors [68]. KiVa also provides materials for teachers and other staff and arranges meetings with them to provide step-by-step guidance and instruction for curriculum lessons to ensure the consistency of teachers' behavior and maintain program quality [68]. Teachers have the opportunity to design their lessons using the program's manual [68]. A guide for parents is designed to provide information about the different forms of bullying and recommendations for prevention when the problem is reported. It also encourages parents to work with the school and teachers to create an effective anti-bullying culture [69]. Visible symbols, logos, imprints on teachers' vests or shirts, and posters are used to make it clear that the school is a KiVa school and bullying incidents will not be tolerated [68].

Williford et al. (2013) undertook a study involving Finnish students in grades 4 to 9, aiming to ascertain the impact of the KiVa intervention on cyberbullying. The study indicated that the influence of the KiVa intervention on post-test cyberbullying outcomes was influenced by students' age. The intervention was notably effective for students aged below 12 years ($p < 0.01$) [69]. For more studies, refer to Table 1.

### 4.2. Olweus Bullying Prevention Program (OBPP)

Based on the idea that bullying should not be part of a child's natural environment, OBPP is one of the most studied and successful anti-bullying efforts in the world [18,70]. The program was originally designed for school children to control violence in schools, but later evolved and expanded to control youth aggression in online settings as well [71]. Like other successful interventions, baseline information is collected to target the program and tailor the interventions for the individual as per needs assessment [72].

The OBPP begins by changing the behaviors of adults in the school to show affection and interest in students' lives, promote rules and regulations against violent behavior, and present themselves as positive role models [73]. In addition, school staff are trained and held accountable for monitoring "hot spots" in order to intervene immediately in bullying behaviors [71]. Identifying bullying incidents and counseling perpetrators, victims, and

their parents through serious talks are also part of the staff training [18]. Group discussions and school staff meetings become regular practice after the implementation of OBPP to reflect on bullying and related prevention efforts at the school [18]. For children and youth training, teachers are encouraged to hold regular class meetings to express and remind goals and ground rules through activities such as role playing, small and large group activities, and discussions [18]. The meeting topics are decided by staff members and serve as an awareness program to illustrate types and subtypes of bullying and promote awareness about respecting others, coping with stress, problem-solving, and using consistent positive and negative consequences [18]. A Bullying Prevention Coordinating Committee (constituting 8 to 15 members from a school) is responsible for all staff training, organizing awareness events, improving school supervision plans, and endorsing school anti-bullying rules [18]. Family nights and after-school leadership programs for adolescents are also designed to involve the community in awareness and prevention programs [18]. The extensive longitudinal investigation conducted by Olweus et al. (2019) involved over 30,000 students in grades 3–11 across 95 schools in central and western Pennsylvania spanning a 3-year period. The study utilized a quasi-experimental extended age-cohort design to examine self-reported instances of bullying behaviors. The outcomes demonstrated that OBPP yielded favorable results in decreasing all forms of bullying, whether experiencing it or perpetrating it ($p < 0.05$) [71]. Similarly, Bowland's short-term study (2011) exhibited a statistically significant decrease in the prevalence of bullying ($p = 0.022$) and instances of peer exclusion ($p = 0.009$) among 7th grade female students. However, there was variability in the statistical outcomes for 8th grade females, and no significant findings emerged for males [74]. OBPP has been most successful in long-term studies. However, a shorter version of the program was found to be less effective (Refer to Table 1).

*4.3. MARC Anti-Bullying Program*

The Massachusetts Aggression Reduction Center (MARC) has developed a schoolwide anti-bullying program to raise awareness and provide solutions to children's social problems, with a focus on bullying and cyberbullying, and to create an overall nurturing school environment [75]. The MARC intervention begins with basic information to customize the program and continues to evolve and improve through ongoing research [76]. The elements of the program are complementary to Olweus' interventions, developing opportunities for teachers to increase awareness, knowledge, concepts, and practical interventions to address bullying and cyberbullying [76]. In addition, MARC includes training a lead trainer in the staff training component and students who are considered authority figures and high-level peers, and these continue to train colleagues and other staff and help younger students in accordance with the program content of MARC [76]. To raise awareness among parents and the community, presentations are developed that provide practical and concrete knowledge about how to eradicate the problem and useful strategies to help adults talk to their children about the phenomenon, ask schools for help, and assist school administrators in successfully resolving bullying situations [76]. In addition, presentations and campaigns led by trained older students continue to reinforce awareness raising for younger children. Program elements for students are accompanied by guides for teachers and parents to reiterate key points and encourage classroom discussions [76]. MARC also holds annual quizzes and contests where students present posters, write poems, create public service announcements, etc. to encourage and recognize positive student behaviors [76]. In addition, MARC curricula are available free of charge to students internationally, which has led to many success stories of the MARC anti-bullying program [76]. The study by McCoy et al. (2018) involving 6th and 7th graders from a middle school in Massachusetts revealed that the qualitative program under investigation received primarily favorable feedback. The students acquired fresh knowledge and became effectively motivated. Students resonated most with the comprehensive insight they gained into digital behaviors and cyberbullying. The practical and straightforward advice presented was highlighted as one of the most beneficial components of the program [76].



However, because cultural differences may make it difficult to implement the positive elements of the program, it is recommended that extensive preparation be undertaken to implement the program in other cultures [77].

### 4.4. ConRed Program

The Knowing, Building, and Living Together on the Internet Program [Conocer, Construir y Convivir en la Red, ConRed] was primarily designed to cope with cyberbullying and its consequences, incorporating psycho-educational research into key intervention strategies for dealing with bullying behaviors [78,79]. Although the ConRed program is based on a holistic school-based approach, the most important target group is students who are technically trained, along with improving their communication and social skills in the online world [80]. The design of the program was based on the assumption that strategies to tackle traditional bullying can be effectively utilized in preventing cyberbullying. The program adopts the Theory of Normative Social Behavior (TNSB), which has proven successful in behavior modification [81], explaining that social behaviors are particularly inferred from peer group intimidation and are heavily influenced by perceived social conventions regarding online behaviors, which are expressed in the form of frequent uploading of personal information and images and constant connection to the virtual world [82].

The three main components of the ConRed program are working on internet addiction, bullying, and empathy [80]. ConRed intervention experts also work with each school's climate planning team for three months to improve perceived control over information available on the Internet, reduce time spent on digital devices, and prevent cyberbullying [80]. The program also involves the implementation of clear policies to combat risks associated with the internet and online social networking, with a special focus on fostering empathy [83]. The main themes of topics covered in the training session include awareness of the internet and social networks, their advantages and risks, and strategies to address online bullying [80]. The program is also based on reflection sessions with quizzes to stimulate consolidation of the acquired knowledge. Like other successful interventions, this program starts with preconceived notions of teachers, parents, and students and ends with a reflection quiz to obtain feedback [80]. Up until now, the program has yielded favorable outcomes and holds promise for application in multicultural environments [80,82]. In a secondary school setting (average age = 13.8 years), Ortega-Ruiz et al. (2012) conducted a study in Spain which disclosed that there was a notable reduction in internet addiction ($p < 0.05$) as well as cyberbullying levels ($p < 0.01$) among male participants. Both boys and girls experienced a decrease in victimization ($p < 0.05$) [80]. In a separate investigation by Del Rey et al. (2016) involving secondary school students aged between 11 and 19 years, it was determined that the ConRed program effectively lessened cyber aggression among male students ($p = 0.04$), although its effectiveness was comparatively lower for female students. In terms of cyber victimization, the experimental group exhibited a decrease, particularly noticeable among boys, while an increase was observed among boys in the control group ($p = 0.003$) [82]. These results were attributed to the intervention's feature of recognizing pre-existing notions held by teachers, parents, and students and adapting them to suit the unique requirements of each institutional setting.

### 4.5. TABBY Anti-Bullying Program

The Threat Assessment of Bullying Behaviors Among Youngsters (TABBY) Internet program is based on Ecological Systems Theory and is one of the interventions in which instruction is provided through online media to reduce cyberbullying and increase awareness of cyber risks [84,85]. Teachers are provided information about cyberbullying in comparison to traditional school bullying, the risks associated with cyberbullying and cybervictimization, and skills to identify, prevent, and address cyberbullying and cybervictimization [85,86]. Legal issues related to cyberbullying are also discussed in the training module for teachers [85]. The TABBY toolbox is provided as an additional resource that includes a checklist, brochure, and videos, and its use is also explained in detail [85]. Parent

seminars are also organized to raise parents' awareness of the issue and provide them with strategies for intervening and preventing risky behaviors [85]. The sessions with the students are based on group brainstorming sessions in which the differences and similarities between jokes, cyberbullying, and aggression are shared and understood by the students [85]. The video sessions are used as stimuli for discussions about students' roles in the virtual world, which is expected to lead to the development of rules and strategies for safe online behaviors [85]. Eventually, the new rules and strategies are shared with the entire school and become part of the school's cyberbullying policy [85]. The outcomes of the TABBY intervention among students aged 13–14 years in Greek secondary schools indicated a decrease in risky behaviors related to cyber activities in the post-intervention results. However, there was no statistically significant difference observed in the post-intervention data between the control and experimental groups ($p = 1.99$) [87]. While Tabby is already a comprehensive program, incorporating a baseline survey can further enhance its effectiveness in implementing interventions within cross-cultural and social contexts. By incorporating the feature of recognizing pre-existing notions held by teachers, parents, and students, and then adapting the program accordingly to meet the specific needs of each institutional setting, it will become better equipped to address cross-cultural differences and tailor interventions accordingly. This additional step can greatly contribute to its suitability in diverse cultural settings.

### 4.6. Cyber Friendly Schools

Cyber Friendly Schools is based on Social Ecological Theory, and its components are developed by incorporating young people's opinions and suggestions to address technological needs and their consequences [88]. To create a positive school environment and combat a bullying culture, strategies are used that equip schools with knowledge about cyberbullying and tactics to support students' emotional and social development [89]. In addition, strategies are developed to strengthen links between schools, homes, communities, and sanctions for cyberbullying practices [89]. Further, student "cyber leaders" are recruited and trained to support staff and other students against bullying and victimization, based on the assumption that teens have a greater awareness of technology and online behaviors than adults [89]. School project teams are provided with resources, including a brochure and student activity booklet, to gather and consolidate basic information about cyberbullying, consequences, legal action, common student online activities, and bystander impact. The booklet also details strategies for school staff to deal with cyberbullying situations [89]. Finally, newsletters discuss in detail updating social media friends lists, students' digital reputation, cybersecurity, and legal issues, which is one of the unique features of this intervention [89].

Teaching and learning resources with different types of activities and online quizzes are provided to receive and provide information to increase their understanding and skills to address bullying [89]. The evaluation of the intervention applied to students aged between 13 and 15 years yielded mixed results in terms of effectiveness, as teachers implemented it poorly due to lack of time and additional time spent on regular school activities [90]. Cyber victimization decreased from years 1 to 2 ($p = 0.034$), but stability was maintained between years 2 and 3 ($p = 0.193$). Conversely, perpetration declined from years 1 to 2 and then to 3. However, the significant negative trend was only statistically significant during the period between the second and third data collection points ($p = 0.006$) [90].

### 4.7. Learning Together

Learning Together is a U.K school-based intervention aimed at improving young people's health and wellbeing, using an innovative whole-school corrective approach that aims to prevent or resolve conflicts between students and staff and prevent bullying to minimize the harm associated with such problems [91,92]. It also provides an opportunity for victims to report and share their feeling with teachers and obtain guidance. Learning

Together consists of staff training in restorative practice, convening and facilitating a school action group, and a social and emotional skills curriculum for students. Learning Together applied to secondary school students for three years had small but significant effects on bullying (control group, mean bullying = 0.34, SE = 0.02 versus experimental group, mean bullying = 0.29, SE = 0.02), which could be important for public health, but it had no effect on aggression (SE > 0.05) [89]. This is an emerging curriculum and needs evidence-based research trials to validate the outcomes. Moreover, the addition of components to handle cyberbullying issues, using baseline survey information, adopting a theory-based approach, and involving teachers and youth in developing or modifying the curriculum are also recommended.

*4.8. No Trap*

No trap is a web-based, online, peer-led approach based on Ecological Systems Theory [93]. The program is delivered in the form of teacher and peer group manuals that serve as resources for their training, in addition to web-based and Facebook information pages [93]. Teachers first receive basic training to raise awareness and intervene [93]. Teachers then actively participate in classroom activities conducted by peer groups and assist their peers in implementing the program with each group of students [93]. During student training, psychologists address these issues by conducting awareness sessions to promote empathy and sensitivity, the role of bystanders, and practical skills through video, discussion, and role playing [93]. Peer leaders receive in-depth training to improve their listening skills and learn how to respond to victims when approached by their peers for help [93]. The final part is equipping leaders with problem-solving and coping strategies to problems [93]. This program has been successfully tested and yielded robust results in a study wherein peer leaders (mean age = 14 years) were volunteers (victimization: $B = 0.025$; SE = 0.005; $p < 0.001$; bullying: $B = 0.017$; SE = 0.004; $p < 0.001$), but undesirable results when the peer leader was appointed by classmates (victimization: $B = -0.000$; SE = 0.006; $p = 0.958$; bullying: $B = 0.005$; SE = 0.005; $p = 0.250$) [94]. It is advisable to delve deeper into the factors contributing to the lack of success in peer interventions when students are responsible for appointing their peers. This method, which allows students to exercise autonomy and the right to vote, is widely favored and considered an effective means of selecting student representatives, aligning with democratic principles. Therefore, a thorough exploration of the reasons behind its failure would be beneficial.

**Table 1.** Comparison of the interventions studied.

| Intervention | Country of Program Development | Country of Program Implementation | Theory/Concept | Duration | Baseline Survey-Needs Assessment | Targeted Outcome Variable | Approach * | Evaluation |
|---|---|---|---|---|---|---|---|---|
| **Interventions Based on Teacher Professional Development** | | | | | | | | |
| PEACE (Preparation, Education, Action, Coping, Evaluation) Pack https://www.flinders.edu.au/research/peace-pack-phillip-slee (accessed on 22 August 2023) | Australia | Australia, Italy, Greece, Japan, Malta, Canada | Social Constructivism | 6 h | No | Traditional bullying (potentially adaptable for cyberbullying) | Proactive | Promising results [28,30–32] |
| ViSC (Viennese Social Competence Program) http://www.viscprogram.eu/ (accessed on 22 August 2023) | Austria | Austria, Germany Kosovo, Cyprus, Romania, Turkiye | Social Ecological Theory [95] | 4 modules in two semesters and several in-school workshops for teachers | Yes | Traditional bullying (potentially modifiable for cyberbullying | Proactive | Promising results [37–39], partially successful [96] |
| Relationships to Grow (RPC) | Italy | Italy | Resilience and Social Exclusion | 6 h of teacher training | No | Cyberbullying | Proactive | Successful in creating awareness but insignificant reduction in cyberbullying rates [41] |
| Media Heroes | Germany | Germany, Austria, Colombia | Theory of Planned Behavior [97] | Long version: 15 sessions (at least 45 min each) Short version: 4 sessions (90 min each) | No | Cyberbullying, but has shown to be effective against traditional bullying as well | Proactive | Promising results [42,43,45,98] |
| Asegúrate | Spain | Spain | Theory of Normative Social Behavior, self-regulation skills, principles of constructivist methodologies | 8 training sessions for teachers | No, but teachers have possibility to tailor the program according to their needs | Cyberbullying | Proactive | Promising results [47], partially successful [48] |

**Table 1.** *Cont.*

| Intervention | Country of Program Development | Country of Program Implementation | Theory/Concept | Duration | Baseline Survey-Needs Assessment | Targeted Outcome Variable | Approach * | Evaluation |
|---|---|---|---|---|---|---|---|---|
| **Interventions Based on Individual Training** | | | | | | | | |
| Stand Alone | The Netherlands | The Netherlands | Not reported | 3 months | Baseline information is replaced with mind mapping technique | Cyberbullying | Reactive | Promising results [54] |
| Cyberbullying Sensitization Program (CBSP) | India | India | Not Reported | 30 h | No | Cyberbullying | Proactive | Promising results [57] |
| Informational Motivational Behavioral Skills (IMB) | United Kingdom-South Africa | South Africa | Health Behavior Change Framework [58] | 50 min | Yes | Cyberbullying | Proactive | Successful, but with very small effect size [59] |
| Prev@cib Anti-bullying Program | Spain | Spain | Ecological Model, Empowerment Theory, Personal and Social Responsibility Model [64,95,99] | 10 sessions (1 h each) | No | Traditional bullying and cyberbullying | Proactive | Successful in local context [61] |
| **Interventions Based on Whole-School Approach** | | | | | | | | |
| KiVA Anti bullying Program https://www.kivaprogram.net/ (accessed on 22 August 2023) | Finland | Finland, United Kingdom, New Zealand, Spain, Italy, Estonia, Belgium, etc. | Social Ecological Theory [95] | 2 days in school, training for teachers with follow-up sessions at university | No, but teachers can tailor the program to their needs | Traditional bullying but modified and applied for cyberbullying as well | Proactive and Reactive | Promising results [68,100–104], successful with modest effect size [69] |
| OLWEUS Bullying Prevention Program (OBPP) https://olweus.sites.clemson.edu/ (accessed on 22 August 2023) | Norway | USA, England, Germany | Not mentioned | A continuous training program with variations in durations for committee, staff, students, parents. | Yes | Traditional bullying but modified and applied for cyberbullying as well. | Proactive | Promising results [71,72,105], mixed results in short-term studies [74] |
| (Massachusetts Aggression Reduction Center) MARC Anti-Bullying Program https://www.marccenter.org/ (accessed on 22 August 2023) | USA | USA | Not Mentioned | Several components with variations in duration for staff, parents, peer leaders, lead trainers, students, etc. | Yes | Traditional bullying and cyberbullying | Proactive and Reactive | Promising results [76] |

**Table 1.** *Cont.*

| Intervention | Country of Program Development | Country of Program Implementation | Theory/Concept | Duration | Baseline Survey-Needs Assessment | Targeted Outcome Variable | Approach * | Evaluation |
|---|---|---|---|---|---|---|---|---|
| ConRed Cyberbullying Intervention Program | Spain | Spain | Theory of Normative Social Behavior [106] | 3-month period, external experts conducted 8 training sessions with students, 2 with teachers, and 1 with families | Yes | Cyberbullying | Proactive | Promising results [80,82] |
| Threat Assessment of *Bullying* Behavior among Youngsters (TABBY) Improved Prevention and Intervention *Program* (TIPIP) | Italy | Italy, Greece | Ecological Systems Theory [95], Threat Assessment Approach | 12 h of teacher training, followed by sessions for parents and students | No | Cyberbullying | Proactive | Mixed results [87] |
| Learning Together https://www.learning-together.eu/bullying-and-cyberbullying/ (accessed on 22 August 2023) | United Kingdom | United Kingdom | Not Mentioned | Not mentioned | No | Traditional bullying | Proactive and Reactive | Significant results in bullying prevention [91] |
| Cyber Friendly Schools https://friendlyschools.com.au/ (accessed on 22 August 2023) | Australia | Australia | Social Ecological Theory [95] | Several components with variations in duration for staff, parents, peer leaders, lead trainers, students, etc. | No | Cyberbullying | Proactive | Successful for the first year but unsustainable in later years [89,90] |
| No Trap | Italy | Italy | Ecological Systems Theory [95] | 4 months | No | School bullying and cyberbullying | Proactive and Reactive | Promising results [107,108], mixed results [92] |

* Proactive interventions aim to prevent bullying by taking preemptive actions, while reactive interventions focus on responding to reported or identified bullying cases.

### 5. Conclusions and Discussion

The exponential growth of social media platforms and versatile online games has opened up avenues for expressing aggression and negative emotions through cyberbullying, particularly when anonymity is preserved. In order to address these behaviors, this discussion focuses on existing effective interventions that have been extensively tested, as well as emerging interventions. Objectives, underlying theories, success rates, and to some extent, strengths, limitations, and suggestions for overcoming those limitations are examined. The ultimate aim is to provide recommendations for implementing these programs and suggest potential improvements.

#### 5.1. Encountering Traditional Bullying and Cyberbullying

Traditional bullying programs have been developed and evaluated positively. According to Slonje et al. (2013), such programs can also address cyberbullying, including the implementation of school anti-bullying policies and engaging students in curriculum-based activities [5]. As a result of these conclusions, many bullying prevention programs have been upgraded to prevent cyberbullying [109] and work together to control emerging problems (refer to column 3, row 2, Table 2). Cyberbullying coexists with traditional bullying, and studies have shown that controlling one form of bullying can lead to other forms of bullying being committed by the perpetrator [110–112]. According to Kowalski et al. (2014), cyberbullying often occurs at the same time as traditional bullying, which implies that cyberbullying is more common in institutions where traditional bullying is more prevalent [112]. This assessment led to the conclusion that both forms of bullying should be addressed in intervention designs.

#### 5.2. Strong Theoretical Framework

Theory is an essential part of scientific research and a quality theory is one that is testable, falsifiable, and parsimonious [113]. A meta-analysis by Tanrikulu (2018) found that few prevention efforts were specifically designed using a strong theoretical approach and most had no conceptual background, raising questions about which components work against bullying and why [9]. The current evaluation also found that some of the underlying theories of successfully implemented and evaluated programs were not well defined or explained, despite years of work to address bullying and its consequences (refer to row 10 in Table 2).

#### 5.3. Baseline Information and Needs Assessment

Baseline assessments are important to serve as a benchmark for measuring project success or failure and establishing priority areas. Baseline information helps stakeholders decide which aspects of a project need more focus [114]. To address cross-cultural differences and practices, some programs began with baseline information to tailor and modify the interventions based on a needs assessments (refer to row 3, column 3 in Table 2). It is recommended that interventions be tailored to needs and begin with baseline information to sustainably address bullying. Similarly, it is not recommended to replicate and apply interventions in cross-cultural studies before they have been adapted to meet the needs of the specific population.

#### 5.4. Unique Content of the Programs

The main theme of most of the interventions was to develop emotional skills, empathy, and awareness of victims, create a positive environment, and provide support with skills to counter bullying. However, some of the programs had additional components, such as working with intercultural skills (ViSC), online communication (ConRed), or sexting and cyber grooming (Prev@cib). These sensitive online behaviors are considered serious and criminal in nature and require more understanding and expanded education to protect children from online risks. When examining antisocial behavior, technology usage emerges as a significant factor that warrants consideration. Memmedova and Selahattin (2018)

highlighted a link between frequent technology use and persistent anxiety issues [115], while Chung et al. (2019) found that anxiety is often accompanied by aggression and related behaviors [116]. Nevertheless, the interventions primarily focused on addressing bullying behaviors tended to prioritize personal grooming. However, only a limited number of interventions, such as the ConRed Program, specifically addressed technology use. It is advisable to thoroughly examine and explore the frequency of online activities as an additional component in addressing and managing bullying behaviors, alongside other relevant factors.

### 5.5. Use of Web-Based Training, Computer Games, and Online Support

Technology-based bullying awareness and prevention interventions have shown remarkable results in reducing bullying [117–119]. Similarly, the use of technological resources such as games, weblinks, and online resources used in previous interventions (refer to rows 5 and 6 in Table 2) can be adapted and adjusted in other interventions that involve face-to-face sessions or a blended approach. However, the effectiveness of these programs needs further research to support these changes.

### 5.6. Reactive/Proactive Approaches

Interventions designed to address bullying behaviors are categorized as proactive when actions are taken to prevent bullying behaviors from occurring [120]. Reactive interventions are those that take actions against bullying once cases are reported or identified to counter the negative consequences of victimization and help victims become psychologically stable through emotional regulation strategies and counseling of the perpetrator [121]. Various interventions operated with distinct components, with some focusing on preventive measures and others on mitigating bullying incidents. However, researchers assert that interventions should encompass elements that address both preventive measures and reactive strategies to manage bullying consequences. Therefore, it is recommended to adopt a combination of techniques that tackle pre-bullying behaviors and effectively address the repercussions of bullying incidents (refer to Table 1 and row 7 in Table 2).

### 5.7. Involving High-Status Peers

Victims are usually reluctant to share their suffering for a variety of reasons, including mistrust of adults and fear of being blamed [16]. They often seek anonymous help in a multitude of ways, such as using online browsers or peer support [50,51]. Unfortunately, only a limited number of interventions focused on involving students as counselors, supporters, and advocates for victims (refer to row 8, column 3 in Table 2). Involving peers and staff members with basic training can be successfully implemented as a reactive approach to victimization.

### 5.8. Empowering Participants

Some anti-bullying programs had unique features and components that could be adopted by other interventions to improve results. One of the unique features found in MARC, ConRed, and Cyber Friendly Schools was the use of quizzes and competitions for encouragement and reinforcement at the end of the interventions. Overall, these interventions have produced profound results [76,82,122]. These kinds of activities along with recognition encourage students, teachers, staff members, and the whole community to actively participate in interventions and create an anti-bullying environment. Similarly, the involvement of young people and teachers in the design of rules, policies, activities, and strategies and the provision of opportunities to modify the existing content according to needs will encourage them to be more active and provide satisfaction by giving them importance in decision making and achieving the goal of the intervention. Asegúrate, OBPP, KiVA, CSBP, Cyber Friendly Schools, and Prev@cib are interventions that focus on one or the other aspect of participants' involvement. Other interventions could also encourage implementers to achieve the goal of the intervention.

### 5.9. Use of Cyberbullying Recognition and Zero Tolerance Features

Continuous reinforcement through landmarks, as employed in KiVA, is a unique feature that reminds bullies of zero tolerance and victims of safety and empowerment. The use of "hot spots" in the OBPP program was also presented as a successful contribution to reducing bullying and victimization. Similar attempts to train teachers and parents to monitor "hot spots" in the cyber world are also suggested for controlling cyber-related risky behaviors.

**Table 2.** Strengths of the interventions.

| Strengths | Supporting Literature | Interventions with Suggested Strengths |
|---|---|---|
| **Working with both traditional bullying and cyberbullying**<br>Maintenance of peer relationships online and offline cannot be separated; therefore, cyberbullying cannot be solved apart from face-to-face interaction. Interventions should address both forms of bullying; otherwise, there have been studies showing that suppressing one form of bullying allows the perpetrator to engage in another form of bullying. | [112,123]<br><br>[110,111] | P.E.A.C.E, ViSC, KiVa, OBPP, Prv@cib, MARC, No Trap, Stand Alone |
| **Modified according to baseline information**<br>Baseline assessments are important to act as a benchmark for measuring project success or failure and establishing priority areas. Thus, it is recommended that interventions should be tailored according to needs. | [114] | OBPP, MARC, ViSC, Stand Alone, P.E.A.C.E, ConRed, IMB Model |
| **Training of lead trainers**<br>There are limitations associated with long-term applications of interventions, such as cost, effort, and time. When the lead trainer is also trained, he or she can provide continuous assistance to other faculty members, resulting in sustainable program results. | [76] | MARC, OBPP |
| **Web-based and online resource interventions**<br>Web-based interventions are also considered cost-effective, convenient, easily accessible, can maintain anonymity/privacy, have potential to tailor the program, and can address a large number of people. | [124] | No Trap, Stand Alone and TABBY, Cyber Friendly Schools |
| **Use of computer games**<br>The use of computer games in bullying control interventions has been shown to significantly reduce cyberbullying. Thus, use of computer games is considered an effective method to reduce bullying and victimization. | [118] | KiVa |
| **Components with both reactive and proactive approaches**<br>A proactive approach is practical to eliminate the issue, but providing psychological support to victims is another important aspect that needs more breadth in interventions. It is recommended that a mix of techniques be employed to handle pre-bullying behavior and post-bullying consequences. | [120] | ViSC, KiVa, MARC, OBPP, Learning Together, No Trap |
| **Preparing high- status peers to help victims**<br>Victims usually seek anonymous help in many different ways, including through online browsers and peer support. A reactive approach to handling victimization can be implemented by providing peers with essential training. Positive peer interaction is among the strongest protective factor against being a bully/victim. | [50,51]<br><br>[125] | MARC, Cyber Friendly Schools, No Trap, KiVa, Media Heroes |
| **Components with hands-on activities**<br>In the context of cyber safety education, providing opportunities for students to observe and perform hands-on skills can benefit all types of learners. In order to ensure safe digital media use, practical skills should be part of the training. | [126] | ViSC, Media Heroes, RPC, Asegúrate, Stand Alone, CBSP, IMB, Prev@cib, KiVa, MARC, ConRed, No Trap |
| **Incentives for active participation**<br>Students can be incentivized to participate in activities that might not be of interest to them at first, which allows them to continue participating. | [127] | MARC |
| **Strong theoretical framework**<br>Cyberbullying perpetration is a phenomenon that can be explained by a wide range of social science theories and the majority of the initial work was atheoretical and descriptive in nature. Nevertheless, some interventions align with psycho-social theories that justify certain components. | [128] | P.E.A.C.E, ViSC, RPC, Media Heroes, Asegúrate, IMB, Prev@cib, KiVa, ConRed, TABBY, Cyber Friendly Schools, No Trap |

## 6. Recommendations

Although many interventions have been designed, modified, implemented and evaluated, the fact is that technology is constantly increasing its impact in all areas, which could lead to more cyberbullying behaviors and consequences [129]. As technology evolves, there is a need to continually improve the measures in place to regulate these risky behaviors. When devising anti-bullying intervention programs, it is crucial to consider continuous improvement, updating, and contextualization of the program content as an important factor [33,130,131]

Shachar et al. (2016) reported that aggression can be countered by engaging students in physical and sports activities and by teaching self-control and emotional regulation [132]. In addition to health benefits, physical activity and sports engagement have been found to be effective methods of controlling bullying in many international investigations and interventions [133–135]. Consistent with the suggestions of Siddiqui et al. (2021), students who are more likely to engage in cyberbullying or risky victimization behaviors can be protected from further consequences by engaging them in sports or outdoor activities and reducing time on digital devices [136]. It is recommended that interventions be designed to reduce children's digital engagement and replace it with alternative physical activities to reduce anxiety and other associated risky behaviors [115,137].

The use of mobile applications and virtual reality (VR) as a countermeasure is also suggested by many researchers, but more evidence-based research is needed to determine the effects. For example, the "Shazam" app or "Unmute Daniel" are some of the technology-based interventions designed to create awareness and prevention of bullying [117]. In addition, virtual learning programs that use animated characters to teach youth how to respond to bullying have shown positive and effective results [119]. It is advisable to add these kinds of technology-oriented activities to existing and emerging interventions.

It is recommended that when evaluating the effectiveness of the program, results should not be validated through self-report, but conclusions should be drawn by involving multiple respondents [130].

## 7. Directions for Future Research

The literature has defined many theories used in the past for behavior management, such as Self-Determination Theory, the transtheoretical model, the Fogg behavior model (FBM), etc. Previous studies reported that Self-Determination Theory was only applied in correlational studies recently designed to address bullying [138–140], which shows that it has the potential to be used for designing different components of bullying interventions, specifically regarding counselling bullies to engage in more healthy activities and victims to stand up for themselves. Similarly, authors discussed the transtheoretical model to a limited extent in traditional bullying prevention [70] and in workplace bullying [141], but its comprehensive integration into successful interventions has not yet been evaluated. In order to address new forms of bullying, it is advisable that emerging theories be integrated into the design of interventions and the results monitored.

**Author Contributions:** Conceptualization, S.S.; methodology, S.S.; validation, A.S.-K.; formal analysis, S.S.; investigation, S.S.; resources, S.S and A.S.-K.; writing—original draft preparation, S.S.; writing—review and editing, A.S.-K.; visualization, S.S.; supervision, A.S.-K.; project administration, A.S.-K.; funding acquisition, A.S.-K. All authors have read and agreed to the published version of the manuscript.

**Funding:** The article processing charges are funded by TU Berlin.

**Institutional Review Board Statement:** Not applicable.

**Informed Consent Statement:** Not applicable.

**Data Availability Statement:** All relevant data is cited and referred in the text.

**Acknowledgments:** The researchers acknowledge the support from the German Research Foundation and the Open Access Publication Fund of TU Berlin.

**Conflicts of Interest:** The authors declare no conflict of interest.

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
