# Peer review of "Successful and Emerging Cyberbullying Prevention Programs: A Narrative Review of Seventeen Interventions Applied Worldwide"

_societies, doi:10.3390/soc13090212_

Round 1

Reviewer 1 Report

Good job! It is interesting and a good contribution to the knowledge and comparison of the various program proposals for the intervention and prevention of cyberbullying. The classifications of the programs and the comparison criteria used seem correct to me.

Author Response

Thank you for appreciating our efforts.

Reviewer 2 Report

Thank you for allowing me the opportunity to review this manuscript. Overall, I found it very informative and feel that it could serve as a helpful resource for institutions that are exploring evidence-based interventions to prevent or decrease instances of cyberbullying.

I have a few suggestion:

1. The primary purpose of a figure is to "facilitate readers' understanding of the work" (APA manual page 195 [edition 7]). Figure 1 did not serve this purpose for me. It is not clear what the numbers in brackets represent (are they aligned to the references or do they represent the number of programs related to each descriptor?) and columnal orientation made me spend time trying to navigate the contents of the table rather than organizing the contents of the manuscript. If the three categories (teachers' professional development, individual's trainings, and whole school approach) are factors derived from a statistical process, that should be noted (this does not seem to be the case). 

2. Table 1 is informative, but the formatting makes it difficult to read. If there is a page limit for publication, this table is too bulky. Since this is not a meta-analysis, there is no need to include information on study design. I recommend categorizing by country, name of intervention, cost,  and a link to where the institution can find information in a conventional table. This would act as a helpful resource.

3. Section 3.1. The words "stop online bullies" is repeated several times.

4. line 266: Use "developing" or "under-resourced" instead of "underprivileged countries."

5. Line 275: words are repeated

6.  Line 287-289: Provide a citation

7. Line 305: What findings are being referenced?

8. Line 472: words are repeated. 

9. Line 550: remove the word "it"

10. Table 2 is great! very easy to read and the focus is clear.

11. Line 653: This is the first mention of theoretical frameworks. The validity of the manuscript would be evident sooner if you discussed these frameworks in the Introduction and made references to them throughout. Institutions may select interventions based on a particular heuristic and it would be helpful if these were aligned. 

Author Response

Thank you for appreciating our efforts. Please find attached file for detailed responses.

Reviewer 3 Report

Overall the paper was informative  and well researched.  

The major piece missing was a lack of statistical information for any of the programs. What statistics did the studies provide?  For example, when you are discusssing the differnet programs, words like - significant reduction; successful and sustainable results; fostering online behavior, etc. don't tell me, as the reader of the document, what those words or phrases mean?  Is this how the authors referred to their results? I would think they included the actual statistical data? This data was needed for each program to give the reader specific knowledge about the program.  

Defne - Traditional bullying as used in your paper.

The other topic missing is critical thinking/analysis of all the programs.  How do they compare and contrast? You did a rudimentary comparison in Table 2 but more analysis was needed. What were the strengths and weaknesses for each program.  What is each program missing?  Are these programs' statistical analysis adequate, very good, or poor. Statistically, school anti-bullying programs are only about 20% effective in stopping bullying. How do these programs compare? Is 20% good enough? Overall which program offers the best outcome?  Why?  Will the country's culture make a difference in how the program was taught?  Whether the program met it's goal in each culture?  what differences were there in the different ccountries?  Did the authors examine that variable?

Additially, when describing the programs, the descriptions lacked much depth.  For the most part paragraphs listed the topics covered and activities used in the education of sudents and teachers.  . Though you mention which countries were involved in various studies in Table 2, it would have been helful for the countries to be listed when you are referring to "countries" in the text. 

You  refer the reader to table 2 when discussing various issues in your program discription.  Table 2 contains a large amount useful information that was a wonderful representation of the various programs  It would be helpful if you gave a bit more identifying information so the reader knows where on Table 2 you are refeerring them to.  Does Table 2 require labels within the table?  For example, each program mght be assigned a letter with each column of the program including a number A1, B3 etc.

Your English is good however you have some unnecessary words (as many of us do) in some of your sentences , for example: (the following is one of the last sentences in your paper)

My editing of this paragraph is imposisible to get throughContinuous reinforcement by the use of (delete)by using recognition features what are these?, as employed (delete) as in KiVA, is  613 (what  is this number for?). is one of the unique features observed that will help (delete) ( to help remind bullies of zero tolerance and 614 victims of safety and empowerment.( don't understand this last portion of the sentence)  I have rewitten the paragraph below.   Because I don't know what recognition features are, my rewrite may nor fit exactally., and I don't understand 614 victims of safety and empowerment- it didn't make sense to me, so that part of the sentence is missing in my edit.

Continuouse reinforcement by using recognition features, as in KiVa, is a unique feature observed to help remind bullies of a zero tolerance and...

You refer to the students as youth, adolescents, students, and children.  This is confusing - children and adolescents are not the same, for example, so as the reader, I didn't know what age group the program studies were using.  It would be beneficial to try to stick to one of the labels or to use  what age group was used in each study of the program.

Author Response

(The authors gave the same response as above.)
